# Policy Stitching: Learning Transferable Robot Policies

Pingcheng Jian[1]   Easop Lee[1]   Zachary Bell[2]   Michael M. Zavlanos[1]   Boyuan Chen[1]

[1]Duke University  [2]Air Force Research Laboratory

**generalroboticslab.com/PolicyStitching**

**Abstract:** Training robots with reinforcement learning (RL) typically involves heavy interactions with the environment, and the acquired skills are often sensitive to changes in task environments and robot kinematics. Transfer RL aims to leverage previous knowledge to accelerate learning of new tasks or new body configurations. However, existing methods struggle to generalize to novel robot-task combinations and scale to realistic tasks due to complex architecture design or strong regularization that limits the capacity of the learned policy. We propose Policy Stitching, a novel framework that facilitates robot transfer learning for novel combinations of robots and tasks. Our key idea is to apply modular policy design and align the latent representations between the modular interfaces. Our method allows direct stitching of the robot and task modules trained separately to form a new policy for fast adaptation. Our simulated and real-world experiments on various 3D manipulation tasks demonstrate the superior zero-shot and few-shot transfer learning performances of our method.

**Keywords:** robot transfer learning, policy stitching

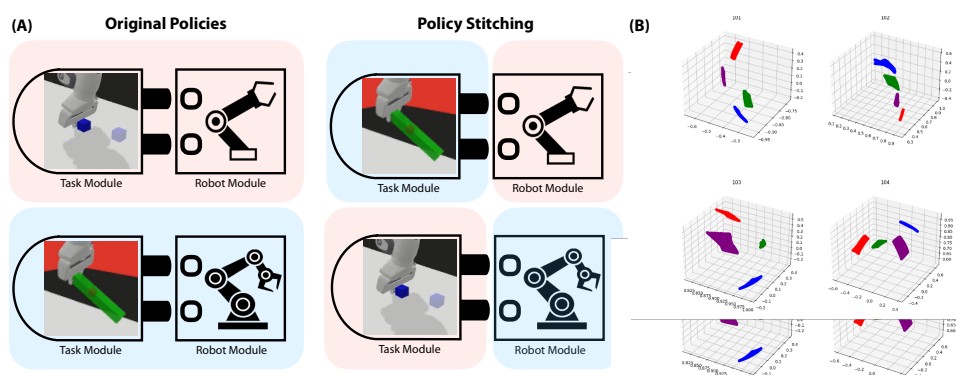

Fig. 1: **Policy Stitching.** (A) Our framework facilitates robot transfer learning among novel combinations of robots and tasks by decoupling and stitching robot and task modules. (B) Motivation example: A robot arm is trained to reach goals in four different target regions using the modular policy. Results from separate training runs with different random seeds (101-104) show misaligned latent representations.

## 1   Introduction

Robots are typically trained to excel at specific tasks such as relocating objects or navigating to predetermined locations. However, such robots need to be retrained from scratch when faced with new tasks or body changes. In contrast, humans demonstrate remarkable capabilities [1] to continuously acquire new skills by drawing on past experiences. Even under physical constraints imposed by injuries, we can still rapidly adapt to perform new tasks. Despite significant advancements in robotics and machine learning, robots still cannot generalize their experience across a wide range of tasks and body configurations.

Model-based robot learning, in particular self-modeling or self-identification learning [2, 3, 4, 5, 6, 7, 8, 9, 10], aims to learn a predictive model of the robot's kinematics and dynamics and then employ this model for various downstream tasks through model predictive control. However, the

7th Conference on Robot Learning (CoRL 2023), Atlanta, USA.

learning process needs to be separated into two stages of robot-specific and task-specific learning. On the other hand, model-free reinforcement learning trains policies end to end, but it often has limited transfer learning performance [11]. Existing efforts regularize one large policy network for multi-task learning by learning routing connections to reuse part of the network weights [12] or assigning task-specific sub-networks [13]. However, the capacity of the large policy network grows exponentially as the number of tasks increases.

We introduce **Policy Stitching (PS)**, a model-free learning framework for knowledge transfer among novel robot and task combinations through modular policy design and transferable representation learning (Fig.1 (A)). We explicitly decouple robot-specific representation (e.g., robot kinematics and dynamics) and task-specific representation (e.g., object states) in our policy design to enable the reassembly of both modules for new robot-task combinations. For instance, given one policy trained for a 3-DoF manipulator ⟨robot⟩ to pick up a cube ⬛ and another policy trained for a 5-DoF ⟨robot⟩ manipulator to pick up a stick ▮, if we would like to have the 3-DoF manipulator pick up a stick now (i.e., ⟨robot⟩ + ▮), our method can directly take the robot module in the first policy and stitch it with the task module in the second policy.

While the modular design allows direct stitching, the reassembled policy may not work at all. We find that this is because the output representation from one neural network module does not align with the desired input representation of another module, particularly when modules are trained on different tasks and robot body configurations. Past work in supervised learning [14, 15] has made similar observations. As a motivating example, in Fig.1(B), we show that, even under the same task and robot setup, latent representations from RL policies trained with different random seeds do not align with each other. More interestingly, they exhibit similar isometric transformations as shown in recent work on supervised learning [14, 15]. See Appendix C.1 for additional details of the isometric transformation phenomenon.

To this end, we further propose to generalize the latent representation alignment techniques from supervised learning to reinforcement learning for robot transfer learning. The key idea is to enforce transformation invariances by projecting the intermediate representations into the same latent coordinate system. Unlike supervised learning, RL does not come with human labels to help select anchor coordinates. We propose to use unsupervised clustering of target states to help resolve this gap. Our method produces aligned representations for effective policy stitching. In summary, our contributions are three-fold:

- **Policy Stitching**, a model-free reinforcement learning framework for robot transfer learning among novel robot and task combinations.
- **Modular Policy Design** for robot-specific and task-specific module reassembly. **Representation Alignment** to learn transferable latent space for direct module stitching.
- Demonstration of the clear advantages of our method in both zero-shot and few-shot transfer learning through **simulated and physical 3D manipulation tasks**.

## 2 Related Work

**Robot Transfer Learning.** Transferring robot policies across various environment dynamics or novel tasks is still an open challenge [16, 17, 18, 19]. Past efforts have proposed to transfer different components in reinforcement learning framework, such as value functions [20, 21, 22], rewards [23], experience samples [24], policies [25, 26, 27], parameters [28, 29, 30], and features [31]. In contrast, our method transfers structured network modules of a policy network across novel tasks or novel robot body configurations. Our modular design is similar to previous work [27], but we do not limit the capacity of the latent space to avoid overfitting. Instead, we enforce invariances among the learned latent representations of different modules through feature alignment. Moreover, our work surpasses previous accomplishments in 2D tasks by demonstrating 3D manipulation skills, both in simulation and the real world.

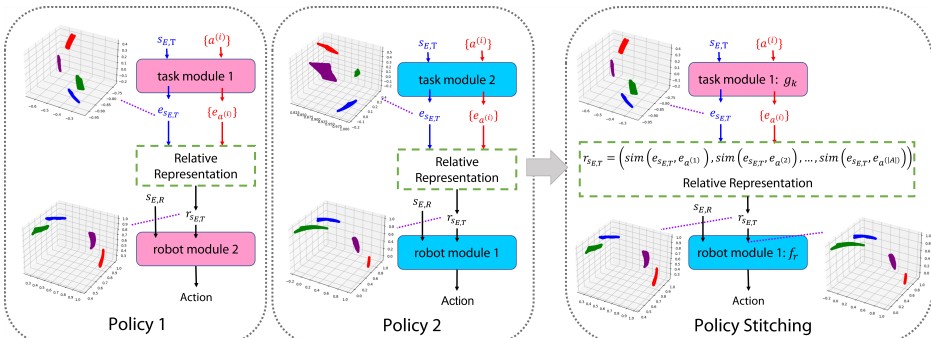

Fig. 2: **Method Overview.** Our modular policy design enables the task module to process task-specific states such as object and environmental states and the robot module to process robot-specific states such as kinematics and dynamics. We generalize relative representations to model-free RL setup to align latent representations from separately trained policies. With both modular design and latent representation alignment, our method allows direct stitching of unseen combination of task and robot module for effective and efficient transfer.

**Meta Learning.** Meta-learning [32, 33, 34, 35, 36, 37] aims to achieve fast adaptation on a new task based on past learning experiences. Recent research has proposed generating better parameter initialization for learning new tasks [30, 38, 39, 40] or using memory-augmented neural networks [41, 42, 43] to assimilate new data swiftly without forgetting the previous knowledge. Another category of methods [44, 45, 46, 47, 48, 49] proposes to use a hypernetwork to generate the parameters of policy networks for different tasks. Our work also aims at fast adaptation to novel robot and task combinations, but our method reuses structured components in policy networks.

**Compositional Reinforcement Learning.** Functional compositional RL has been used in zero-shot transfer learning [27], multi-task learning [12], and lifelong learning [13]. Some approaches learn the structure of the modules [50, 51, 12, 13], but these trained modules can only be functional in a large modular system and cannot work when stitched with new modules. Another method [27] tries to make the network module reusable, but the module alignment problem prevents it from working on 3D tasks. Our work can directly stitch policy modules to improve zero-shot and few-shot learning performances in simulated and physical 3D manipulation tasks.

## 3 Method: Policy Stitching

Policy Stitching focuses on enabling effective knowledge transfer between different tasks or robot bodies. Our method consists of two main components: the modular policy design and transferable representations achieved through latent space alignment. Our framework is designed to be compatible with various model-free RL algorithms.

### 3.1 Modular Policy Design

We propose modular policy design to decompose the policy into two distinct modules focusing on robot specific and task specific information. Our design provides a straightforward ingredient for module reuse and stitching. Consider an environment, $E$, with a robot $r$ and a presented task $k$, we formulate the problem as a Markov Decision Process with an observed state $s_E$ and action $a_E$. We denote the learning policy as $\pi_{E_{rk}}(a_E \mid s_E)$ parameterized by a function $\phi_{E_{rk}}(s_E)$. We utilize the model-free RL formulation, specifically the Soft Actor-Critic (SAC) [52] algorithm.

We decompose the state $s_E$, the policy function $\phi_{E_{rk}}(s_E)$, and the Q-function into a **robot-specific** module and a **task-specific** module (Fig. 2). The state $s_E$ is decomposed into a robot-specific state $s_{E,\mathcal{R}}$ and a task-specific state $s_{E,\mathcal{T}}$. The robot-specific state $s_{E,\mathcal{R}}$ only consists of the joint angles of the robot, while the task-specific state $s_{E,\mathcal{T}}$ includes task information at hand such as the current position and orientation, linear and angular velocity, and goal position of the object. Similarly, the policy function $\phi_{E_{rk}}(s_E)$ is also decomposed into a task-specific module $g_k$ to encode the input task states, and a robot-specific module $f_r$ to implicitly capture robot kinematics and dynamics from input robot states. We follow the same design to decompose the Q-function into task-specific module $q_k$

and robot-specific module $h_r$. Note that this decomposition is similar to previous work [27] with generalization on both actor and critic network. Formally, our policy function and Q-function can be re-written as follows:

$$\phi_{E_{rk}}(s_E) = \phi_{E_{rk}}(s_{E,\mathcal{T}}, s_{E,\mathcal{R}}) = f_r(g_k(s_{E,\mathcal{T}}), s_{E,\mathcal{R}}), \tag{1}$$

$$Q_{E_{rk}}(s_E, a_E) = Q_{E_{rk}}(s_{E,\mathcal{T}}, s_{E,\mathcal{R}}, a_E) = h_r(q_k(s_{E,\mathcal{T}}), s_{E,\mathcal{R}}, a_E). \tag{2}$$

For two policy networks $f_{r1}(g_{k1}(s_{E,\mathcal{T}}), s_{E,\mathcal{R}})$ and $f_{r2}(g_{k2}(s_{E,\mathcal{T}}), s_{E,\mathcal{R}})$, we define Policy Stitching as constructing another policy network $f_{r2}(g_{k1}(s_{E,\mathcal{T}}), s_{E,\mathcal{R}})$ by initializing the task module with parameters from $g_{k1}$ and initializing the robot module with parameters from $f_{r2}$. The Q-function is stitched in similar way.

We name the modules after their main functionalities. Such modular design does not completely separate the information processing from each module. Since we train the entire policy end to end, the gradients flow in between the two modules during backpropagation. However, by explicitly enforcing their main functionalities through our modular design, we can still obtain effective transfer learning by stitching modules to leverage previously acquired knowledge. After the stitching, we can either use the new policy for zero-shot execution or perform few-shot learning for fast adaptation.

### 3.2 Transferable Representations through Latent Space Alignment

Our modular policy design allows for direct stitching, but a simple stitching approach may not yield optimal performance. Previous research [27] attributes the issue as overfitting to a particular robot and task, and has proposed to use dropout and very small bottleneck dimensions. However, the capacity of the policy is largely limited to simple 2D tasks and low-dimensional task states. Instead, we identify the fundamental issue as the lack of alignment enforcement of the latent embedding space at the output/input interface of two modules. Even under the same robot-task combination, the latent embedding vectors between the robot and task modules from different training seeds exhibit an almost isometric transformation relationship (Fig.1(B)). While similar observation has been made in supervised learning [15, 53], we have identified and addressed this issue in the context of RL and demonstrate the effectiveness in knowledge transfer across novel robot-task combinations.

We propose to generalize Relative Representations [15] in supervised learning to model-free RL. Unlike other techniques in supervised learning [54, 14, 55], Relative Representations does not introduce additional learnable parameters. The key idea is to project all latent representations at the interface between two modules to a shared coordinate system among multiple policy networks. Through this invariance enforcement of transformation, we can establish a consistent latent representation of the task-state that aligns with the desired latent input of the robot module.

However, the original Relative Representations relies on the ground-truth labels in the supervised dataset to provide the anchor points for building the latent coordinate system. Our setup in RL does not come with such labels. The anchor points are analogous to the concept of basis in a linear system, hence should be as dissimilar from each other as possible. To overcome this challenge, we first collect a task state set $\mathbb{S}$ by rolling out two trained naive policies in their own training environments. We then perform unsupervised clustering with k-means [56] on these task states to select an anchor set $\mathbb{A}$. Specifically, we select $k$ anchor states that are closest to the centroids of the clusters. The number of clusters $k$ depends on the dimension of the latent representation.

We want to represent every embedded task state $e_{s^{(i)}} = g_k(s^{(i)})$ with respect to the embedded anchor states $e_{a^{(j)}} = g_k(a^{(i)})$, where $a^{(j)} \in \mathbb{A}$ and $g_k$ is the task-specific module that embeds the task states. To this end, we capture the relationship between an embedded task state $e_{s^{(i)}}$ and an embedded anchor state $e_{a^{(j)}}$ using a similarity function $sim: \mathbb{R}^d \times \mathbb{R}^d \to \mathbb{R}$, which calculates a similarity score $r = \text{sim}(e_{s^{(i)}}, e_{a^{(j)}})$. Given the anchor set $a^{(1)}, \ldots, a^{(|\mathbb{A}|)}$, the transferable representation of an input task state $s^{(i)} \in \mathbb{S}$ is calculated by:

$$r_s = \left(\text{sim}\left(e_{s^{(i)}}, e_{a^{(1)}}\right), \text{sim}\left(e_{s^{(i)}}, e_{a^{(2)}}\right), \ldots, \text{sim}\left(e_{s^{(i)}}, e_{a^{(|A|)}}\right)\right), \tag{3}$$

which is a vector of length $|\mathbb{A}|$, and each element is the similarity score between the embedded task state and an embedded anchor state. We intentionally choose the cosine similarity as our similarity

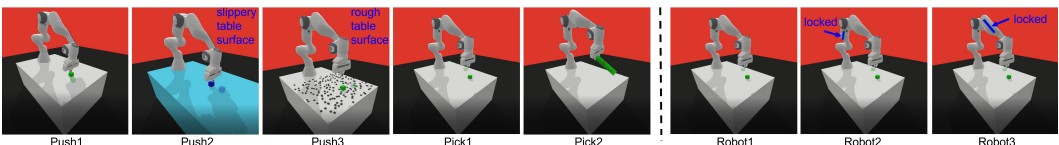

Fig. 3: **Simulation Experimental Setup**: From left to right, we show different task and robot configurations: standard table, slippery table, rough terrain with small rocks, cube object, stick object, 7-DoF Franka Emika Panda arm, third joint locked, and fifth joint locked.

measure because it is invariant to reflection, rotation ,and rescaling. Although it is not invariant to vector translation, we add a normalization layer before calculating cosine similarity to mitigate this issue. In this way, our proposed Relative Representation for RL projects latent representations from different policies into a common coordinate system. Therefore, it overcomes the isometric transformation issue of the original latent representations and makes them better aligned and invariant.

### 3.3   Implementation Details

We challenge our methods in sparse reward setup and use hindsight experience replay [57] to encourage exploration. Both the task and robot modules are represented as MLPs [58] with four layers and 256 hidden dimensions. The last layer of the task module and the first layer of the robot module have a dimension of 128, thus the dimension of the latent representation and the selected anchor set are also 128 ($k = 128$). Detailed network structures of our method and other baselines can be found in Appendix A. Each of our policy is trained on 7 threads of the AMD EPYC 7513 CPU and 1 NVIDIA GeForce RTX 3090 GPU. Each training epoch consists of $35,000$ steps.

## 4   Experiments

We aim to evaluate the performance of Policy Stitching (PS) in both zero-shot and few-shot transfer setups. Furthermore, we generalize our experiments to a physical robot setup to demonstrate the practical applicability and real-world performance. Finally, we provide quantitative and qualitative analysis to understand the role of the learned transferable representations in effective policy stitching.

### 4.1   Simulation Experiment Setup and Baselines

We modify the panda-gym environment [59] to simulate 5 manipulation task environments and 3 distinct robots with varying kinematics (Fig.3). The tasks consist of 3 push scenarios, where the robot must push a cube to the goal position on surfaces with very different friction properties, and 2 pick tasks that involve pick and place task of objects with different shapes. In zero-shot evaluation, the stitched policy is directly tested without fine-tuning and each experiment consists of 200 test trajectories, which are repeated 5 times to report the mean and standard deviation. In few-shot evaluation, the stitched policy first interacts with the environment for additional 10 epochs to store data in the replay buffer and is then fine-tuned for a few epochs with SAC. We train with 3 random seeds to report the mean and standard deviation.

**Robot-Task Combination.** We evaluate PS on 8 robot-task combinations to cover policy stitching in both similar and dissimilar setup. We list all combinations as the titles of sub-figures in Fig.4. For instance, "t: Pi1-R1" represents a task module trained in the Pick1 task with Robot1, while "r: Pu1-R2" represents a robot module trained in the Push1 task with Robot2. These modules are then stitched together to form a new combination that has not been jointly trained together.

**Baselines.** We compare our method to the approach of Devin et al. [27], which uses a similar modular policy but with small bottleneck and dropout as regularization to alleviate the modules misalignment issue. In the few-shot test, we also include an additional comparison to increase the bottleneck dimension of the Devin et al. [27]'s method to the same dimension as our method, since the original bottleneck dimension is too small to capture necessary information for complex 3D tasks. Furthermore, we provide an ablation study with our modular policy design but no latent

|     | Touching Rate (%) | | | | Success Rate (%) | | | |
| --- | --- | --- | --- | --- | --- | --- | --- | --- |
|     | *PS* | *PS(Ablation)* | *Devin et al.* | *Plain* | *PS* | *PS(Ablation)* | *Devin et al.* | *Plain* |
| E1 | $73.0 \pm 1.4$ | $\mathbf{75.8 \pm 4.4}$ | $44.8 \pm 3.1$ | $0.0 \pm 0.0$ | $\mathbf{26.9 \pm 3.6}$ | $13.9 \pm 3.5$ | $11.7 \pm 2.6$ | $6.5 \pm 2.5$ |
| E2 | $\mathbf{99.9 \pm 0.2}$ | $93.5 \pm 2.0$ | $78.3 \pm 5.4$ | $17.9 \pm 2.7$ | $\mathbf{24.4 \pm 1.5}$ | $6.8 \pm 0.7$ | $10.0 \pm 0.9$ | $9.3 \pm 2.1$ |
| E3 | $90.8 \pm 2.3$ | $\mathbf{95.6 \pm 0.8}$ | $82.6 \pm 1.6$ | $0.0 \pm 0.0$ | $\mathbf{16.5 \pm 1.9}$ | $7.2 \pm 1.5$ | $9.8 \pm 2.5$ | $8.4 \pm 3.7$ |
| E4 | $\mathbf{95.4 \pm 1.9}$ | $88.7 \pm 1.1$ | $49.6 \pm 1.1$ | $1.6 \pm 1.0$ | $13.2 \pm 1.1$ | $\mathbf{14.9 \pm 1.0}$ | $11.8 \pm 2.4$ | $9.0 \pm 1.4$ |
| E5 | $\mathbf{14.7 \pm 2.7}$ | $9.6 \pm 1.6$ | $8.6 \pm 0.3$ | $2.8 \pm 0.9$ | $2.1 \pm 0.4$ | $\mathbf{4.0 \pm 1.9}$ | $2.9 \pm 1.0$ | $3.8 \pm 1.6$ |
| E6 | $55.6 \pm 2.4$ | $\mathbf{80.8 \pm 3.4}$ | $30.5 \pm 4.2$ | $0.1 \pm 0.2$ | $8.8 \pm 1.9$ | $\mathbf{11.6 \pm 2.0}$ | $10.5 \pm 2.1$ | $9.0 \pm 0.7$ |
| E7 | $\mathbf{41.8 \pm 1.7}$ | $13.1 \pm 2.9$ | $9.8 \pm 1.3$ | $0.0 \pm 0.0$ | $\mathbf{4.7 \pm 1.6}$ | $3.6 \pm 0.8$ | $3.0 \pm 1.6$ | $3.2 \pm 1.4$ |
| E8 | $18.4 \pm 1.3$ | $\mathbf{18.8 \pm 2.6}$ | $13.6 \pm 1.1$ | $0.0 \pm 0.0$ | $\mathbf{5.3 \pm 1.7}$ | $3.0 \pm 0.7$ | $3.1 \pm 0.5$ | $3.0 \pm 0.7$ |

Tab. 1: **Success rates and touching rates of zero-shot transfer**

representation alignment to study its importance. We also build a Plain baseline which is an MLP that takes in all the states at the first layer and has no modular structure. When transferring to novel task-robot combination, we split the Plain MLP network into two parts (i.e.,top-half and bottom-half as in Appendix Fig. 8(c)) and perform the same reassembling operation as the modular networks. Specifically, when performing the stitching operation of the Plain baseline, the top half of a Plain network is stitched to the bottom half of another Plain network. This operation that preserves a sub-network from the old task and adds a new sub-network for the new task has been widely used in other works [60, 61, 62, 63, 64]. See Appendix A for architecture details.

**Metrics.** We use success rate (task completion within a fixed number of steps) and touching rate (contact with the object during the task) as our evaluation metrics. In challenging transfer scenarios (e.g., a robot trained for a pushing task is required to perform a picking task), all methods may exhibit low success rates. Therefore, we use touching rate as a more informative metric to indicate whether the robot is engaging in meaningful interactions rather than arbitrary movements. Effective touching behavior can result in improved exploration for few-shot transfer learning, since touching serves as a preliminary behavior for picking or pushing.

#### 4.1.1 Results: Zero-Shot Transfer in Simulation

E1-E3 are easier robot-task combinations than E4-E8 since the unseen setup is closer to their original combinations. As shown in Tab.1, for E1-E3, PS achieves significantly better or comparable results than Devin et al. [27] and the PS(Ablation) method, suggesting the latent representation alignment serves as the fundamental step to enable effective module stitching. Our method also outperforms Plain MLP policy, which indicates both modular policy design and latent representation alignment are beneficial.

For more challenging cases (E4-E8) where both task and robot configurations are drastically different, though all methods do not exhibit strong success rates, PS demonstrates notably higher touching rates. This implies more meaningful interactions that can potentially lead to a higher success rate

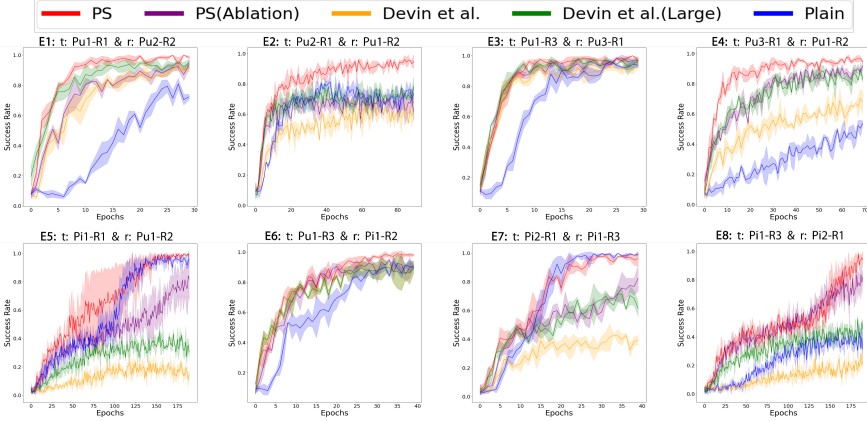

Fig. 4: **Success rates of the few-shot transfer learning in simulation.**

and faster adaption in our few-shot experiments. Qualitatively, Plain baseline shows limited attempts to complete the task with frequent aimless arm swinging. Though PS(Ablation) and Devin et al. [27] methods show higher touching rates than the Plain baseline, they lack consistent attempts to move the target object to the goal after touching. In contrast, our PS method makes the robot attempt to push the object to the goal in most experiments. We show such qualitative comparisons in our supplementary video.

### 4.1.2 Results: Few-shot Transfer in Simulation

In Fig.4, PS achieves the highest or comparable success rates. Notably, in E2 and E4, while other methods tend to get stuck at local minima and reach a plateau at sub-optimal success rates, PS converges to significantly higher rates. In some experiments such as E1, E4, E5 and E7, PS transfers much faster and achieve higher success rates during the early phase of fine-tuning. This high transfer effi-

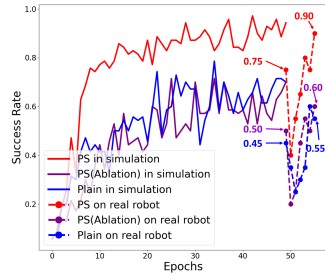

Fig. 5: **Real World Setup**.

ciency supports our hypothesis in the zero-shot experiment that the high touching rates achieved by PS can facilitate transfer learning through meaningful interactions. Even in challenging scenarios where all methods struggle to achieve a high success rate, PS can quickly adapt based on limited interactions. Furthermore, while some methods may achieve comparable results in a few instances, PS is the only method that demonstrates consistent satisfactory performance across all scenarios, suggesting that PS serves as a stable transfer learning framework.

### 4.2 Real World Experiment

First, we aim to evaluate the zero-shot transfer performance of PS in the real world by directly transferring the simulated policies to novel robot-task combinations. Moreover, we are interested in accessing the feasibility of continuously improving such policies given limited real-world interactions.

**Setup.** In our real-world experiment (Fig.5), we include a Robot1 as a 6-DoF UR5 arm, a Robot2 as the same UR5 arm but with the fifth joint being locked, a Task1 as pushing a cube to a goal position, and a Task2 as pushing a cylinder. The friction coefficient between the object and the table is 0.16 for Task 1 and 0.72 for Task 2. We first train Robot1-Task2 and Robot2-Task1 pairs in simulation. We then create Task1-Robot1 through policy stitching. For the few-shot experiments, we first fine-tune the policy for 50 epochs in simulation and test it in the real world. Following this, we then allow the policy to interact with the physical world for an additional 6 epochs (5 hours). We use an Intel RealSense D435i to detect the ArUco markers [65] attached to the object and the goal position on the table. Both objects are 3D printed. The cylinder has a sandpaper at its bottom to increase the friction coefficient. All evaluation results are based on 20 testing instances.

Fig. 6: **Few-Shot Results**. Solid curves show the fine-tuning in simulation. Dotted curves show the further fine-tuning in the real world.

**Zero-Shot Results.** PS achieves a success rate of **40%** and a touching rate of **100%** in zero-shot transfer. In comparison, both the PS(Ablation) and the Plain baseline have 0% success rate. The PS(Ablation) achieves a 75% touching rate, while the Plain baseline cannot even touch the object. The superior performances of PS indicate the great potential of policy stitching to generalize to novel robot-task combinations in real-world settings.

**Few-Shot Results.** As shown by the solid curves in Fig.6, after the first stage of fine-tuning in simulation, PS achieves the highest success rate of about 90% in simulation, surpassing both the PS(Ablation) and Plain methods (∼70%). When this policy is directly transferred to the real UR5 arm, PS still achieves the highest success rate at 75%, followed by the PS(Ablation) at 50% and the Plain method at 45%.

These results indicate that the modular design with relative representation achieves a promising success rate on a new task in both simulation and the real world after a few epochs of fine-tuning. Due

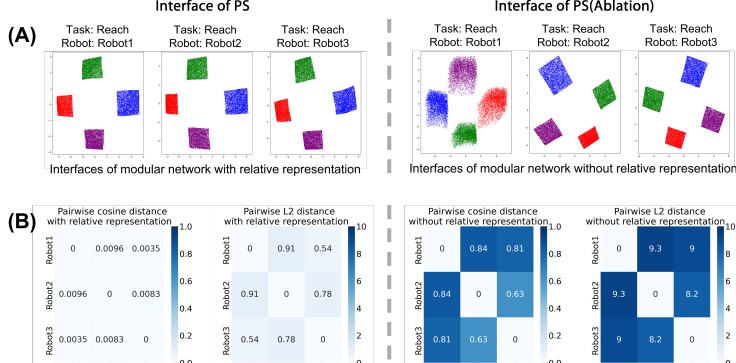

Fig. 7: **Latent Space Analysis.** We train robots with 3 different kinematics for the reaching task and visualize their latent space colorized by 4 groups of task states, similar to Fig.1. (A) Visualization of the 2D principal components of the latent representations. (B) The average pairwise cosine distances and L2 distances between each pair of the stitched latent states.

to the sim2real gap, the success rates of all methods drop to some extent. However, after 6 epochs of fine-tuning on the real robot platform shown as the dotted curves, PS results in a final success rate of 90%. Although the success rates of both baselines (i.e., 60% and 55%) are improved after the fine-tuning, there is still a significant gap with PS. Through both zero-shot and few-shot experiments, we ensure that our method performs effectively, efficiently, and reliably in physical scenarios.

### 4.3    Analysis: Latent Space at Module Interface

To understand how our latent space alignment benefits effective policy stitching by learning transferable representations, we provide further quantitative and qualitative analysis. Since the dimension of the latent embedding at the task-robot module interface is 128, we cannot directly visualize the latent representations. We apply the Principal Component Analysis (PCA) [66] to reduce the dimension of the representation to 2. We train robots with 3 different kinematics on a reaching task with and without our latent space alignment. We then colorize the principal components based on their task states and group them into four directions as in our motivation example (Fig.1(C)). As shown in Fig.7(A), the latent spaces of all policies trained with PS remain consistent across various training environments, while others without latent space alignment still show approximate isometric transformations. The consistent latent space explains the effectiveness of our policy stitching method.

To account for potential information loss during PCA visualization, we also provide quantitative analysis. In Fig.7(B), we calculate the pairwise cosine distances and L2 distances (Appendix C.2) between the raw high-dimensional latent states of different stitched policies. The pairwise distances are significantly smaller when the latent representations are aligned. PS achieves an average cosine distance of 0.0055 and L2 distance of 0.240, while ablation has a much higher cosine distance of 0.865 and L2 distance of 1.275. This analysis indicates that the latent representations of normal modular networks differ greatly from each other, and the latent representation alignment can successfully reduce such differences and facilitate the learning of transferable representations for policy stitching.

## 5    Conclusion, Limitation and Future Work

In this work, we propose Policy Stitching, a novel framework to transfer robot learning policy to novel robot-task combinations. Through both simulated and real-world 3D robot manipulation experiments, we demonstrate the significant benefits of our modular policy design and latent space alignment in PS. PS paves a promising direction for effective and efficient robot policy transfer under the end-to-end model-free RL framework.

One limitation is that our current method of selecting anchor states based on clusters of test states may not generalize to scenarios with high-dimensional state representations, such as images. An exciting future direction is to study self-supervised methods to disentangle latent features for anchor selections. This can further help generalize PS to more complex tasks and reward settings. Another limitation is that our adapted relative representation method still cannot enforce strict invariance of the latent features, hence requires some level of fine-tuning. It will be interesting to explore alternative methods for aligning modules of networks without the need for anchor states. We also plan to explore various robot platforms with different morphology to generalize PS to more diverse platforms.

**Acknowledgments**

This work is supported in part by ARL under awards W911NF2320182 and W911NF2220113, by AFOSR under award #FA9550-19-1-0169, and by NSF under award CNS-1932011.

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

# Appendix

## A   Network Structure

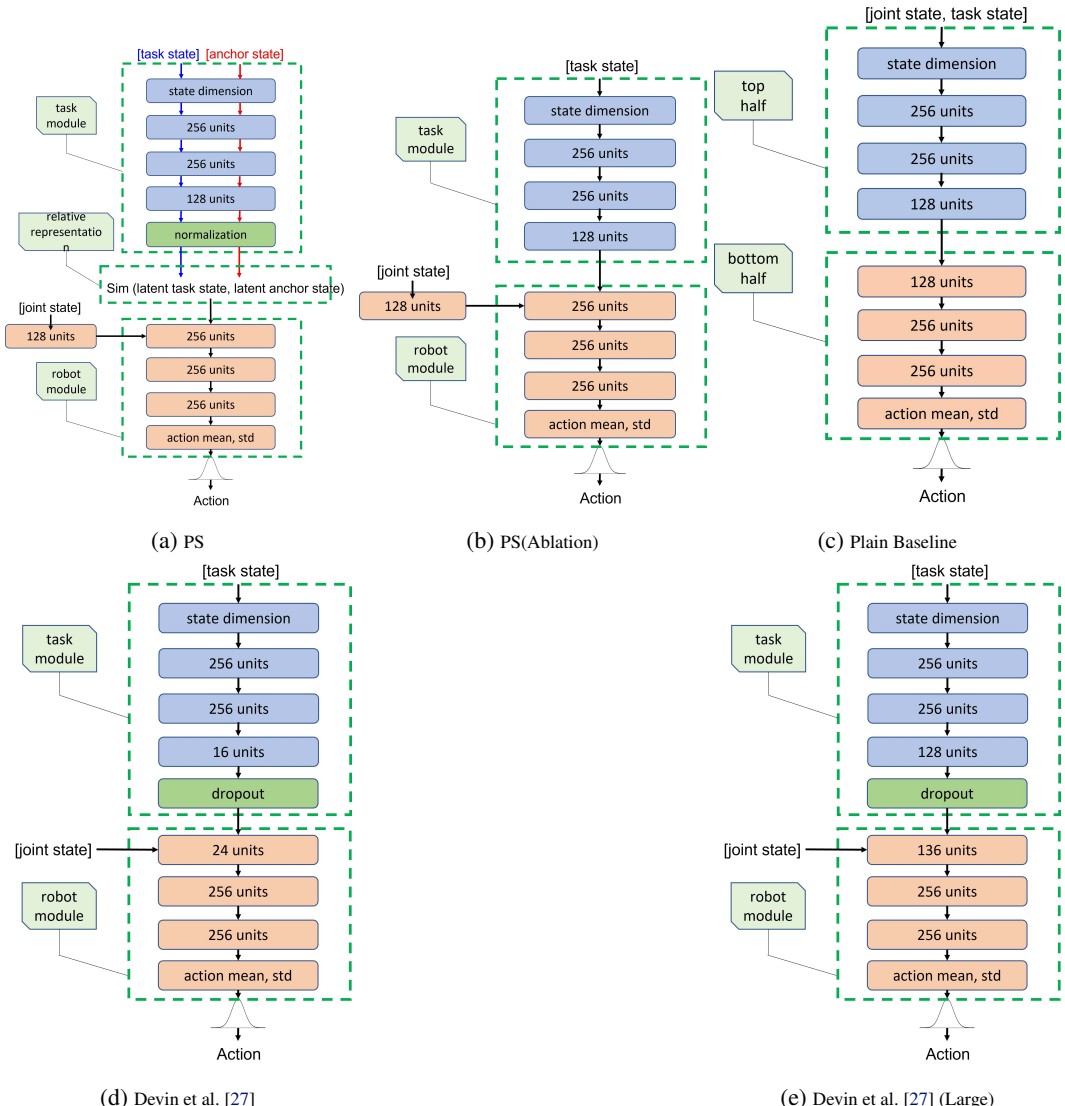

Fig. 8: Detailed network structure of PS, PS(Ablation), Devin et al., Devin et al.(Large) and Plain Baseline methods.

# B Policy Stitching Algorithm

---

**Algorithm 1** Zero-shot Transfer with Policy Stitching

---

**Train Plain Policies in Source Environments:**
Train Plain policy 1 $\pi_{r1k1}$ with SAC in the environment $E_{r1k1}$.
Train Plain policy 2 $\pi_{r2k2}$ with SAC in the environment $E_{r2k2}$.

**Collect Anchor States:**
Roll out $\pi_{r1k1}$ in $E_{r1k1}$ to gather a set of task states $\mathbb{S}_1$
Roll out $\pi_{r2k2}$ in $E_{r2k2}$ to gather a set of task states $\mathbb{S}_2$
task state set $\mathbb{S} \leftarrow \{\mathbb{S}_1, \mathbb{S}_2\}$
K-means center set $\mathbb{C} \leftarrow k - means(\mathbb{S})$
Select anchor set $\mathbb{A} = \{a^{(i)}\}$ which are task states closest to the K-means center set $\mathbb{C}$.

**Train Modular Policies in Source Environments:**
Initialize modular policy 1 $\phi_{E_{r1k1}}$ with $g_{k1}$ for its task module and $f_{r1}$ for its robot module.
Initialize modular policy 2 $\phi_{E_{r2k2}}$ with $g_{k2}$ for its task module and $f_{r2}$ for its robot module.
**for** each iteration **do**
    **for** each environment step **do**
        Embed task state $e_{s_t} \leftarrow g_{k1}(s_t)$
        Embed anchor state $e_{a^{(i)}} \leftarrow g_{k1}(a^{(i)})$
        Calculate the relative representation $r_{s_t}$ according to equation (3)
        $action_t \sim f_{r1}(r_{s_t})$
        $s_{t+1} \sim p_{r1k1}(s_{t+1}|s_t, action_t)$
        $\mathcal{D} \leftarrow \mathcal{D} \cup \{(\mathbf{s}_t, action_t, r(\mathbf{s}_t, action_t), \mathbf{s}_{t+1})\}$
    **end for**
    **for** each gradient step **do**
        update $g_{k1}$ and $f_{r1}$ with SAC algorithm
    **end for**
**end for**
**for** each iteration **do**
    **for** each environment step **do**
        Embed task state $e_{s_t} \leftarrow g_{k2}(s_t)$
        Embed anchor state $e_{a^{(i)}} \leftarrow g_{k2}(a^{(i)})$
        Calculate the relative representation $r_{s_t}$ according to equation (3)
        $action_t \sim f_{r2}(r_{s_t})$
        $s_{t+1} \sim p_{r2k2}(s_{t+1}|s_t, action_t)$
        $\mathcal{D} \leftarrow \mathcal{D} \cup \{(\mathbf{s}_t, action_t, r(\mathbf{s}_t, action_t), \mathbf{s}_{t+1})\}$
    **end for**
    **for** each gradient step **do**
        update $g_{k2}$ and $f_{r2}$ with SAC algorithm
    **end for**
**end for**

**Policy Stitching:**
Initialize the task module with parameters from $g_{k1}$.
Initialize the robot module with parameters from $f_{r2}$.
Construct a stitched policy $\phi_{E_{r2k1}}(s_E) = f_{r2}(g_{k1}(s_{E,\mathcal{T}}), s_{E,\mathcal{R}})$.

**Test the Stitched Policy in the Target Environment:**
Roll out the stitched policy $\phi_{E_{r2k1}}(s_E)$.
Calculate the success rate and touching rate.

---

---

**Algorithm 2** Few-shot Transfer Learning with Policy Stitching

---

**Train Plain Policies in Source Environments:**
Train Plain policy 1 $\pi_{r1k1}$ with SAC in the environment $E_{r1k1}$.
Train Plain policy 2 $\pi_{r2k2}$ with SAC in the environment $E_{r2k2}$.

**Collect Anchor States:**
Roll out two trained policies in their own training environments to gather a set of task states $\mathbb{S}$ during these interaction processes.
K-means center set $\mathbb{C} \leftarrow k - means(\mathbb{S})$
Select anchor set $\mathbb{A}$ which are task states closest to the K-means center set $\mathbb{C}$.

**Train Modular Policies in Source Environments:**
Initialize modular policy 1 $\phi_{E_{r1k1}}$ with $g_{k1}$ for its task module and $f_{r1}$ for its robot module.
Initialize modular policy 2 $\phi_{E_{r2k2}}$ with $g_{k2}$ for its task module and $f_{r2}$ for its robot module.
**for** each iteration **do**
    **for** each environment step **do**
        Embed task state $\boldsymbol{e}_{\boldsymbol{s}_t} \leftarrow g_{k1}(\boldsymbol{s}_t)$
        Embed anchor state $\boldsymbol{e}_{\boldsymbol{a}^{(i)}} \leftarrow g_{k1}(\boldsymbol{a}^{(i)})$
        Calculate the relative representation $\boldsymbol{r}_{\boldsymbol{s}_t}$ according to equation (3)
        $action_t \sim f_{r1}(\boldsymbol{r}_{\boldsymbol{s}_t})$
        $\boldsymbol{s}_{t+1} \sim p_{r1k1}(\boldsymbol{s}_{t+1}|\boldsymbol{s}_t, action_t)$
        $\mathcal{D} \leftarrow \mathcal{D} \cup \{(\mathbf{s}_t, action_t, r(\mathbf{s}_t, action_t), \mathbf{s}_{t+1})\}$
    **end for**
    **for** each gradient step **do**
        update $g_{k1}$ and $f_{r1}$ with SAC algorithm
    **end for**
**end for**
**for** each iteration **do**
    **for** each environment step **do**
        Embed task state $\boldsymbol{e}_{\boldsymbol{s}_t} \leftarrow g_{k2}(\boldsymbol{s}_t)$
        Embed anchor state $\boldsymbol{e}_{\boldsymbol{a}^{(i)}} \leftarrow g_{k2}(\boldsymbol{a}^{(i)})$
        Calculate the relative representation $\boldsymbol{r}_{\boldsymbol{s}_t}$ according to equation (3)
        $action_t \sim f_{r2}(\boldsymbol{r}_{\boldsymbol{s}_t})$
        $\boldsymbol{s}_{t+1} \sim p_{r2k2}(\boldsymbol{s}_{t+1}|\boldsymbol{s}_t, action_t)$
        $\mathcal{D} \leftarrow \mathcal{D} \cup \{(\mathbf{s}_t, action_t, r(\mathbf{s}_t, action_t), \mathbf{s}_{t+1})\}$
    **end for**
    **for** each gradient step **do**
        update $g_{k2}$ and $f_{r2}$ with SAC algorithm
    **end for**
**end for**

**Policy Stitching and Q-function Stitching:**
Initialize the policy task module with parameters from $g_{k1}$.
Initialize the policy robot module with parameters from $f_{r2}$.
Construct a stitched policy $\phi_{E_{r2k1}}(s_E) = f_{r2}(g_{k1}(s_{E,\mathcal{T}}), s_{E,\mathcal{R}})$.
Initialize the Q-function task module with parameters from $q_{k1}$.
Initialize the Q-function robot module with parameters from $h_{r2}$.
Construct a stitched Q-function $Q_{E_{r2k1}}(s_E, a_E) = h_{r2}(q_{k1}(s_{E,\mathcal{T}}), s_{E,\mathcal{R}}, a_E)$.

**Few-Shot Transfer Learning:**
Fine-tune the stitched policy $\phi_{E_{r2k1}}(s_E)$ and the stitched Q-function $Q_{E_{r2k1}}(s_E, a_E)$ with SAC in the target environment $E_{r2k1}$.

---

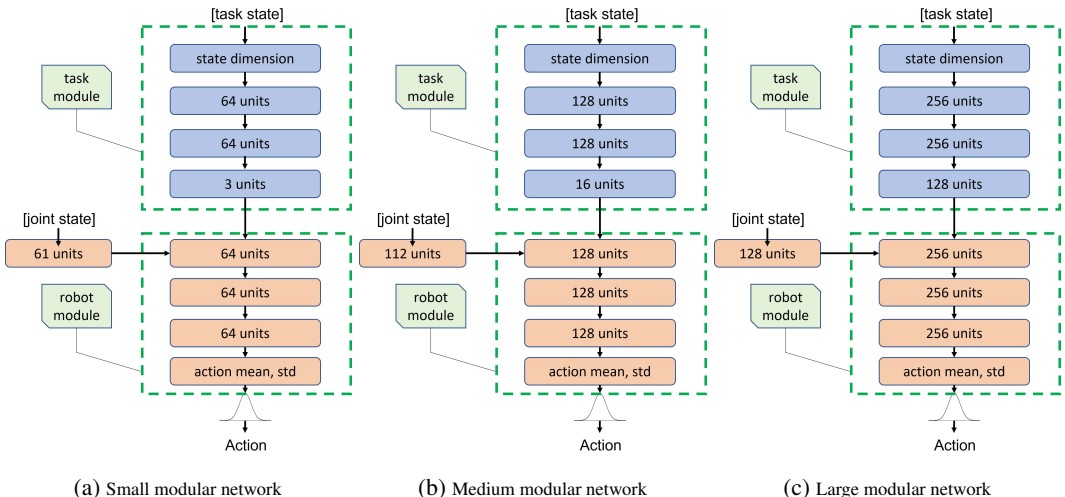

(a) Small modular network     (b) Medium modular network     (c) Large modular network

Fig. 9: The detailed architectures of the modular networks with three different interface dimensions.

## C Analysis of the Module Interface

We carry out additional analysis of the latent representations across different sizes of modular networks. We build three different sizes of modular networks, each interface dimension being 3D, 16D, and 128D as shown in Fig.9. The transferable representation is added to these networks as shown in Fig.8a. We construct six networks (3 different sizes, with and without relative representation) to perform the reaching task as described in Fig.1.

### C.1 Visualization of the latent representations at the modules interface

For the small networks with 3D interfaces, we plot the 3D latent representations directly. For the medium and large networks with 16D and 128D interfaces, we use PCA [66] to reduce the dimension to 2D. Fig.10 shows the visualization of the interface of the six networks trained with different random seeds. The isometric transformation relationship is shown for the PS(Ablation) method across all sizes of modular networks, and with the help of transferable presentation, PS achieves near invariance. Similarly, Fig.11 shows the interface of the networks trained with different robot types. The transferable representation achieves near invariance to isometric transformations across all types of robots.

PCA is an information lossy compression process. The PCA method only guarantees identical output results when the input data sets are identical. When the input data sets are similar but not identical, the output results may vary considerably. In our experiments using PCA for visualization, we have observed that the PCA results of most interfaces with transferable representations are similar. However, in rare cases, we have noticed significant differences in the PCA results. As shown in Fig.12, the original 3D latent states have very similar distributions across the four different runs, but after the dimension reduction to 2D with PCA, the visualization results show isometric transformations. Moreover, in the case of small modular networks with 3D interfaces, achieving a high success rate of approximately $100\%$ often requires a considerable amount of training time. Occasionally, the network may converge to a local minimum with a success rate of around $90\%$. When it converges to a local minimum, the latent representation at its interface typically differs from those that converge to the global minimum. PCA only provides an intuitive idea of the behavior at module interface, thus, we accompany these visualizations with quantitative analysis.

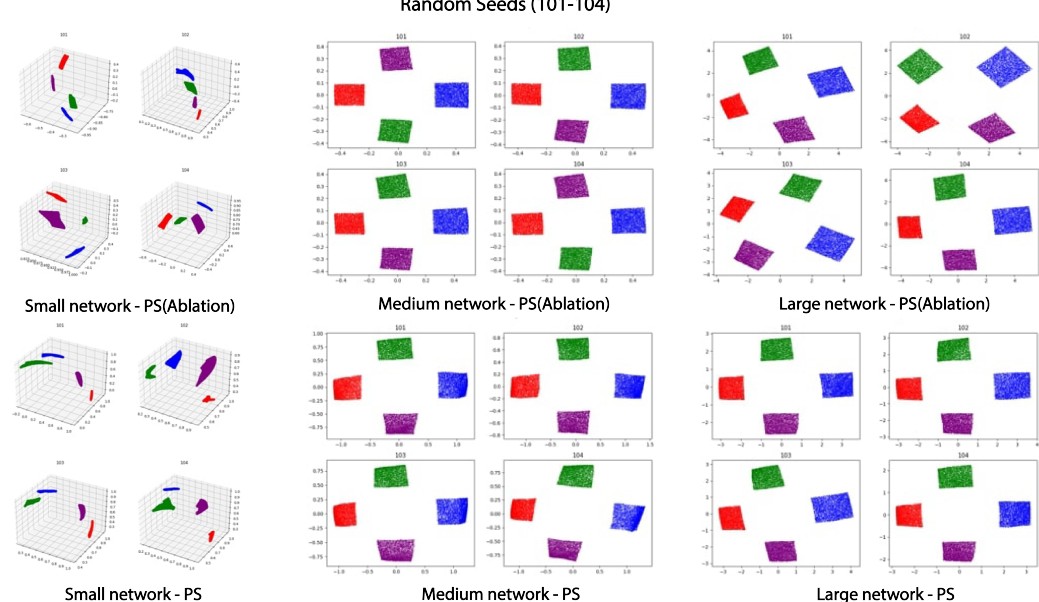

Fig. 10: **Latent Space Visualization** Train each policy network four times with four different random seeds (101-104). Without transferable representation, the latent representations at the interfaces have an approximate isometric transformation relationship. With relative representation, these latent representations are isometrically similar.

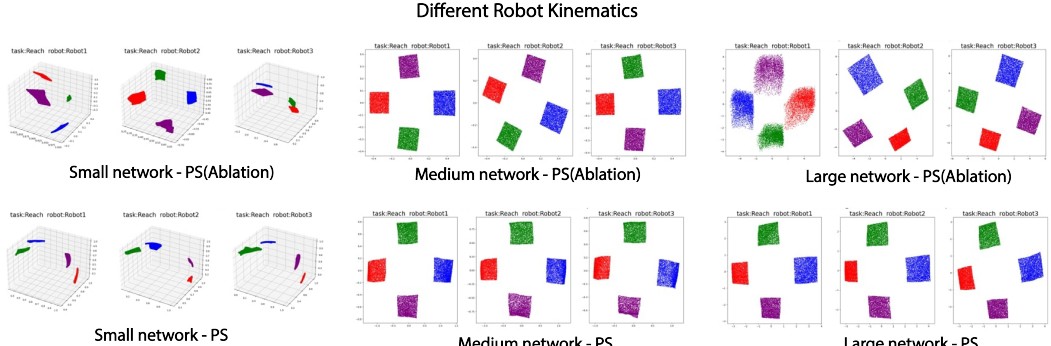

Fig. 11: **Latent Space Visualization** Train each policy network for the reaching task with three different types of robots as shown in Figure 3. With relative representation, the latent representations at the module interfaces are nearly the same across different environments. Without it, they have an approximately isometric transformation relationship.

## C.2 Quantitative analysis of the latent representations at the modules interface

To measure the similarity between two different latent representations, we use cosine and L2 pairwise distances. We compute the pairwise distance between two latent task states derived from the same input state. By considering a dataset of input states, we calculate the mean of the pairwise distances across all input states, obtaining the average pairwise distance between two modular networks.

Given an input task state set $\mathbb{S}_{E,\mathcal{T}}$, the average pairwise cosine distance and L2 distance are defined as

$$\bar{d}_{cos} = \sum_{i=1}^{|\mathbb{S}_{E,\mathcal{T}}|} \left( 1 - S_C \left( g_k^1 \left( s_{E,\mathcal{T}}^i \right), g_k^2 \left( s_{E,\mathcal{T}}^i \right) \right) \right) / \left| \mathbb{S}_{E,\mathcal{T}} \right|, \tag{4}$$

$$\bar{d}_{L2} = \sum_{i=1}^{|\mathbb{S}_{E,\mathcal{T}}|} d_{L2} \left( g_k^1 \left( s_{E,\mathcal{T}}^i \right), g_k^2 \left( s_{E,\mathcal{T}}^i \right) \right) / \left| \mathbb{S}_{E,\mathcal{T}} \right|, \tag{5}$$

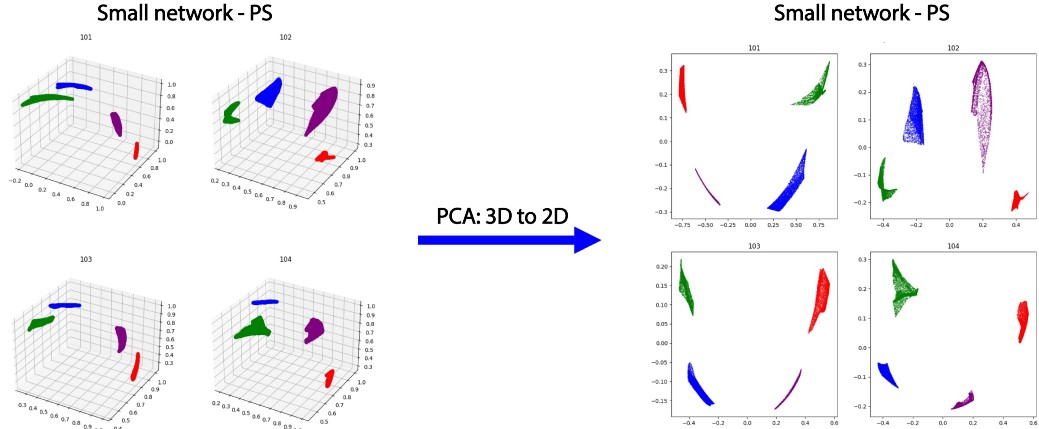

Fig. 12: **Limitation of visualization with PCA.** Raw latent distributions are very similar to each other in the 3D space, but after compression to 2D with PCA, the visualization results are quite different.

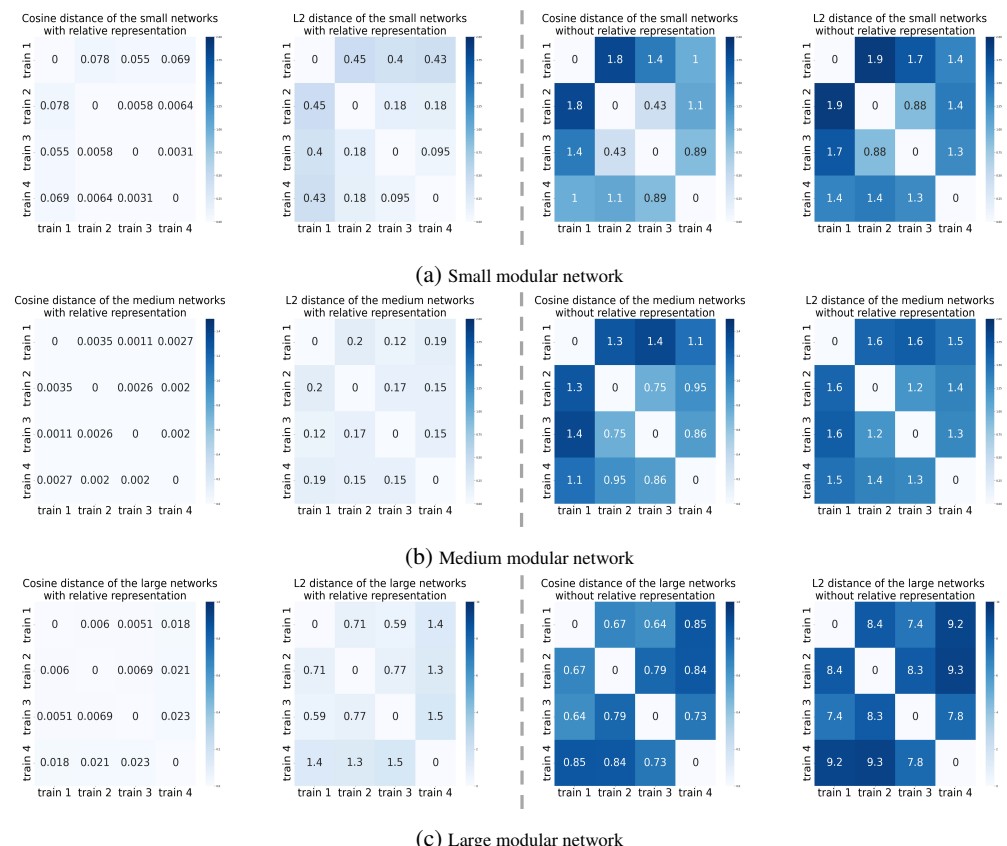

Fig. 13: Cosine and L2 distances of different networks with and without transferable representation for different random seeds.

where $S_C(\boldsymbol{a}, \boldsymbol{b}) = \frac{\boldsymbol{a}\boldsymbol{b}}{\|\boldsymbol{a}\|\|\boldsymbol{b}\|}$ is the cosine similarity and $d_{L2}(p, q) = \|p - q\|$ is the L2 distance. Fig.13 shows the average pairwise distances of modular networks trained with four different random seeds. We calculate the distances for different sizes of networks shown in Figure 9 . We also calculate the mean and standard deviations of the data in Fig.9 and present them in Tab. 2. The results show that the transferable representation largely reduces the average pairwise distances of the latent spaces between different training runs.

We also train the modular networks in different environments and calculate the pairwise distances at the interfaces. Specifically, we train the policy networks on the reaching task with different robots shown in Figure 3. The average pairwise distances are shown in Figure 14 and we calculate the mean values and standard deviations in Tab.3. These quantitative results show that the relative representation makes the module interfaces much more similar to each other when trained in different environments.

|  | cosine distance | L2 distance |
| --- | --- | --- |
| PS | **0.0363 ± 0.0319** | **0.289 ± 0.141** |
| PS(Ablation) | 1.106 ± 0.434 | 1.426 ± 0.322 |

(a) Small modular network

|  | cosine distance | L2 distance |
| --- | --- | --- |
| PS | **0.00231 ± 0.00073** | **0.1633 ± 0.0294** |
| PS(Ablation) | 1.054 ± 0.218 | 1.442 ± 0.151 |

(b) Meidum modular network

|  | cosine distance | L2 distance |
| --- | --- | --- |
| PS | **0.013 ± 0.007** | **1.051 ± 0.368** |
| PS(Ablation) | 0.753 ± 0.081 | 8.386 ± 0.687 |

(c) Large modular network

Tab. 2: Mean and standard deviation values of the average pairwise distances between trainings with four different random seeds (101-104)

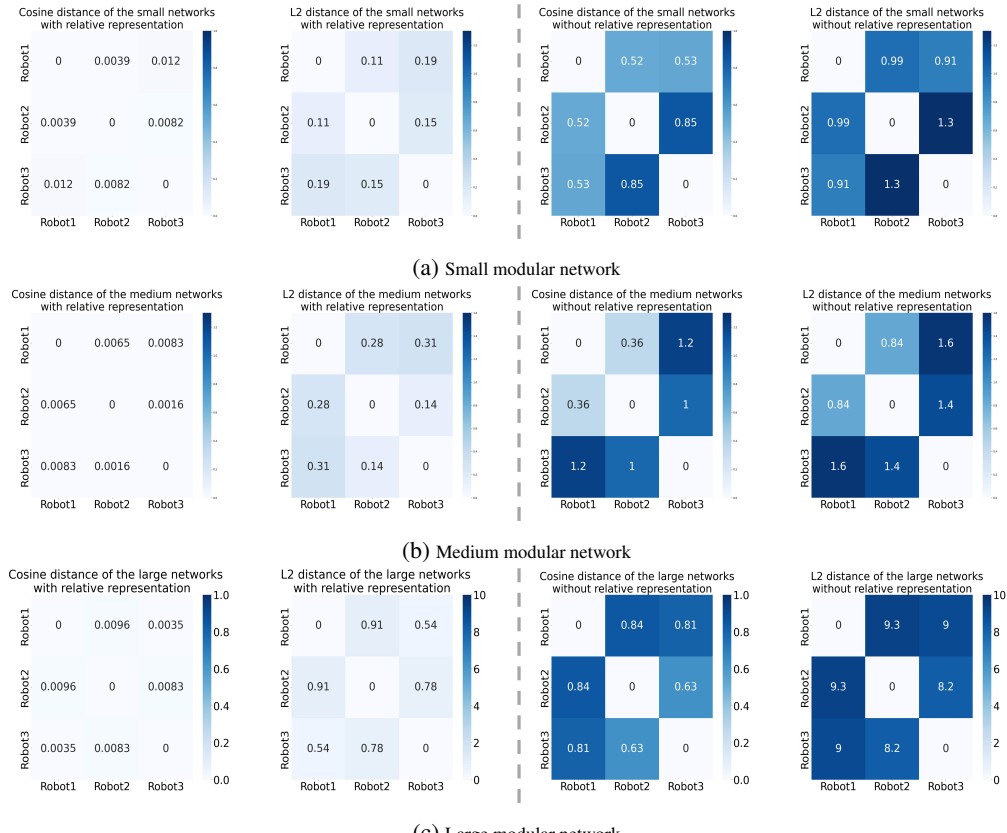

(a) Small modular network

(b) Medium modular network

(c) Large modular network

Fig. 14: Cosine and L2 distances of different networks with and without transferable representation for different robot setup.

|  | cosine distance | L2 distance |
|---|---|---|
| PS | $\mathbf{0.0082 \pm 0.0035}$ | $\mathbf{0.149 \pm 0.032}$ |
| PS(Ablation) | $0.633 \pm 0.153$ | $1.066 \pm 0.165$ |

(a) Small modular network

|  | cosine distance | L2 distance |
|---|---|---|
| PS | $\mathbf{0.0055 \pm 0.0029}$ | $\mathbf{0.240 \pm 0.075}$ |
| PS(Ablation) | $0.865 \pm 0.367$ | $1.275 \pm 0.311$ |

(b) Medium modular network

|  | cosine distance | L2 distance |
|---|---|---|
| PS | $\mathbf{0.0071 \pm 0.0026}$ | $\mathbf{0.743 \pm 0.152}$ |
| PS(Ablation) | $0.760 \pm 0.094$ | $8.822 \pm 0.448$ |

(c) Large modular network

Tab. 3: Mean and standard deviation values of the average pairwise distances between trainings with three different types of robot kinematics

