# OpenReview forum: "Policy Stitching: Learning Transferable Robot Policies"
_robot-learning.org/CoRL/2023/Conference — CoRL 2023 Poster_

### Official Review · Reviewer_2mFg · 2023-07-07

**Confidence:** 4
**Originality:** Very Good
**Technical Quality:** Very Good
**Clarity Of Presentation:** Very Good
**Impact:** 4

**Recommendation:**

Strong Accept: I recommend accepting the paper and will argue for my recommendation even if other reviewers hold a different opinion.

**Review:**

This paper proposes a modular policy design in Policy Stitching, where the policy is divided into two separate modules -
one concentrating on robot-specific information and the other on task-specific information.
This design facilitates easy reuse and combination of modules. The environment is defined with a robot `r` and task `k`
and is formulated as a Markov Decision Process with state `s_E` and action `a_E`. The learning policy `π` is
parameterized by a function $\phi E_{r,k}$. The Soft Actor-Critic (SAC) algorithm is used as model-free RL.
The state `s_E`, policy function $\phi E_{r,k}$., and Q-function are decomposed into robot-specific and task-specific modules.
The robot-specific state `(s_E,R)` includes information like joint angles (i.e. observations that relate to the robotics dynamics), whereas the task-specific state `(s_E,T)` includes information related to the task, such as position, orientation, linear and angular velocity, and goal position of the object.

Equations (1) and 2  show how the `policy function` and `Q-function` can be broken down into task-specific and robot-specific components,
facilitating modular design.  The authors acknowledge that this approach may face issues when combining modules, as the output representation of one module may not align with the input representation of another, particularly when they are trained on different tasks or robot configurations. This issue is exemplified through latent representations from reinforcement learning policies that, even under identical conditions,  do not align due to differences in random seeds during training, and interestingly, exhibit a phenomenon called isometric transformations.

This work further proposes adapting latent representation alignment techniques from supervised learning to reinforcement
learning to address the alignment issue in robot transfer learning through Policy Stitching.
The core concept is to enforce transformation invariances by projecting the intermediate representations
into a common latent coordinate system.  Since the method used here, `Relative Representations` relies on ground-truth labels in supervised datasets to establish anchor points, the authors suggest utilizing the k-means clustering algorithm  to select anchor points from the training set, which serves as the basis for the latent coordinate system.
These anchor points should be as dissimilar as possible and are selected based on their proximity to the centroids of the clusters formed by k-means.

**Experiments**

PS is evaluated in both zero-shot and few-shot transfer setups. The robotics simulation tasks include object picking tasks such as `pick up a cube` and `pick up a stick`. These tasks are ran under sparse reward setup using hindsight experience replay.  Real world experiment are `Task1` : pushing a cube to a goal position, and a `Task2` as pushing a cylinder.

Figure 7 is also interesting where the authors present a latent space analysis where they train robots with three different kinematics for a
reaching task and visualize their latent space, which is color-coded based on four groups of task states. These representations are 2D principal components of the latent representations, providing insights into the structure of the  latent space and `(B)` displays the average pairwise cosine distances and L2 distances between each pair of the stitched latent states. This visualization and analysis characterize effectiveness of the latent representations.

**Baselines**

`PS` is compared with `Devin et al` approach which also uses a modular policy but  employs small bottlenecks and dropout for regularization to tackle module misalignment. Additionally, the authors conduct an ablation study using their modular policy design without
latent representation. They also introduce a baseline model, `Plain`, which is a simple Multi-Layer Perceptron (MLP) without a modular structure.

Additionally,  the authors conduct an ablation study using their modular policy design without latent representation alignment.



**Quality Of The Limitations Section:**

Limitations are addressed clearly

**Questions For Rebuttal:**

- I understand that for the experiments, the tasks are ran under sparse reward setup and with hindsight experience replay. But of course this might limit the choice of tasks that can be ran (object picking are still fairly simplistic experiments). I was wondering what prevents this method to work for more challenging manipulation tasks, perhaps under dense reward setting? I'm wondering in terms of the tasks to solve, how far it can be pushed and whether the authors tried other tasks that failed.
- In terms of terminology, usually in the field, we refer to dynamics vs task related representations. There are many frameworks trying to attempt to learn a decomposed representation for the reward function (assuming a dense reward that already captures environment dynamics and the goals of the task itself) . So I find the terminology used throughout this paper as `robot-specific information` slightly divergent. Even though its accurate, you are explicitly using robot related information such as joints, velocity etc but all these capture the robot dynamics. So I still prefer to stick with the terminology of decomposing dynamics and task (i.e. reward related component).
- How tied is this decomposition to the reward itself? Because as far as I understood, this method tries to capture robot-related and task/goal related observations very separately without considering the reward itself (as its sparse reward in this case, the reward provides no information until task completion).

**Robotics Focus:**

Sufficient demonstration on hardware

**Summary Of Paper:**

This work  introduces a model-free learning framework called `Policy Stitching (PS)`
that facilitates the transfer of knowledge among different combinations of robots and
tasks through a modular policy design and transferable representation learning.
PS distinctly separates robot-specific aspects (like kinematics and dynamics) and
task-specific aspects (like object states) within the policy design.
This separation enables the recombination of these modules for new robot-task pairings.
For example a robot arm's module trained to pick up a cube can be combined with a task module trained for a
different robot to pick up a stick, thus enabling the original robot to pick up a stick by "stitching" these policies together.


**Summary Of Recommendation:**

This paper proposes to learn decomposable dynamics vs tasks representations to solve robotics manipulation tasks. The methodology presented here is novel and practical. Overall the presentation is decent. However I would like to see more challenging robotics tasks, because the capability of this method is only demonstrated on simple pushing tasks.

---

### Official Review · Reviewer_DSoM · 2023-07-18

**Confidence:** 4
**Originality:** Good
**Technical Quality:** Very Good
**Clarity Of Presentation:** Good
**Impact:** 3

**Recommendation:**

Weak Accept: I recommend accepting the paper, but will not argue for my recommendation if the majority of other reviewers have a different opinion.

**Review:**

This paper is overall well-motivated with a broad set of experiments supporting the claims. There are areas for improvement regarding clarity and presentation quality. Please see the specific breakdown of points as follows:

Strengths:
* Studies an important problem (transfer) and introduction motivates the problem well
* Experiment results show meaningful improvements in performance compared to baselines and self-ablations
* Great to see real robot results supporting the simulation results!
* Visualizations of the task representations support the claim that this method learns invariant representations

Weaknesses:
* When presenting the modular policy design (102-119) the authors should acknowledge the prior work of Devin et al., as this design was proposed before in their work. While Devin et al. is acknowledged in the related works and experiments section, their work should be directly acknowledged in this section as well.
* The work of Relative Representations is a significant component in this work, however the authors dedicate relative little writing to explain this work. It would be great to expand on why Relative Representations induces invariant representations, as this was not clear to me immediately.
* Experiments can be more impressive if demonstrating different robots with different morphologies rather than a Panda arm with locked joints
* This is already mentioned in limitations section, but experiments in image domains can help to further strengthen the experiments.
* Further clarification needed on the Plain baseline. What do you mean on line 192 by “perform the same reassembling operation as the modular networks”?

**Quality Of The Limitations Section:**

Limitations are addressed clearly

**Questions For Rebuttal:**

See the list of weaknesses in the section above

**Robotics Focus:**

Sufficient demonstration on hardware

**Summary Of Paper:**

This paper presents Policy Stitching, a method that aims to facilitate transfer to new robots and tasks by learning modular policies. This work builds on the prior work of learning modular policies proposed Devin et al. The paper highlights a major limitation of existing work: the internal task representation is not guaranteed to be invariant to factors such as training seeds. Consequently the paper stresses the importance of learning invariant task representations for facilitating transfer to new settings. To enable this invariance, this paper utilizes the recent work of Relative Representations, by representing the task state as a sequence of similarity scores to anchor states. In experimental evaluations the authors show that the proposed enables better zero-shot and few-shot transfer to a set of tasks with different task and robot designs. Additional quantitative visualizations confirm that the method learns invariant task representations.

**Summary Of Recommendation:**

The paper is well motivated with a good set of experiments. Still experiments have room for growth and writing has room for improvement (see list of weaknesses).

**update after rebuttal** I have carefully reviewed the reviews and the author responses to all reviews. I am still recommending weak accept. For a stronger paper I recommend that the authors study transfer to different morphologies of robots.

---

### Official Review · Reviewer_BozE · 2023-07-19

**Confidence:** 4
**Originality:** Very Good
**Technical Quality:** Very Good
**Clarity Of Presentation:** Very Good
**Impact:** 3

**Recommendation:**

Weak Accept: I recommend accepting the paper, but will not argue for my recommendation if the majority of other reviewers have a different opinion.

**Review:**

Strengths

- The key insights of the paper regarding modular policy design although is not particularly novel, the instantiation with aligning latent representations is simple, insightful, and practical.

- The intuition regarding latent space alignment is very clearly described in the text and in figure 2, and the paper overall is easy to follow.

- The experiments with different robot morphologies, and different task settings are carefully chosen to answer the research questions. In addition, sim2real results demonstrate feasibility of the approach on real robot pushing tasks.

Weaknesses

- The main weakness of the paper is that low-dimensional state information about objects (like position, orientation etc.) are necessary for the current instantiation of the framework, so application to tasks beyond pushing in the real world (e.g. articulated object manipulation) are limited, as it is difficult to quantify object state information.

- Another weakness is that the representations for robot state and task are not "really" separated as it is the same neural network encoding both the information - this is true for the policy network as well as for the q function network. With high dimensional observations, and complex tasks, the current approach might fail because of this, and more careful modular design might be necessary.



**Quality Of The Limitations Section:**

Limitations are addressed clearly

**Questions For Rebuttal:**

- Is it possible to scale the current appraoch to tasks beyond pushing in the real world (e.g. articulated object manipulation) where it is difficult to quantify object state information?

- Another weakness is that the representations for robot state and task are not "really" separated as it is the same neural network encoding both the information - this is true for the policy network as well as for the q function network. With high dimensional observations, and complex tasks, the current approach might fail because of this, and more careful modular design might be necessary. Is it possible to better ensure this modularity or it is actually not necessary to do this even for complex tasks and high-dim observations (e.g. images)?

**Robotics Focus:**

Sufficient demonstration on hardware

**Summary Of Paper:**

This paper proposes an approach for policy stitching, by combining different parts of a policy representing robot state information, and task information for a novel test-time combination. The main insight in the paper is to have a modular policy design and align the latent representations between the modular components. Experiments on different simulated and real robot manipulation tasks show that the proposed policy stitching approach can solve novel robot-task combinations zero-shot, and achieve better performance with few-shot fine-tuning.

**Summary Of Recommendation:**

The paper proposes an interesting formulation of policy stitching, building upon prior works in this domain. Although I have concerns about scaling the approach to high dimensional observations and more complex tasks (since object state information is required), but the current instantiation is still useful for several tasks like pushing, pick and place etc. and so I am recommending weak object.

---

### Official Review · Reviewer_ZQPk · 2023-07-21

**Confidence:** 3
**Originality:** Good
**Technical Quality:** Good
**Clarity Of Presentation:** Poor
**Impact:** 2

**Recommendation:**

Weak Reject: I recommend rejecting the paper, but will not argue for my recommendation if the majority of other reviewers have a different opinion.

**Review:**

Advantage: The paper studies an interesting problem of learning transferable policy, especially for novel combinations of robots and tasks, through modularized design. The problem is valuable, and the modularized idea is promising. It is impressive that the learned policy can be transferred to real-world scenarios.

However, there is still significant room for improvement. The paper’s presentation is rather vague and unclear. For example, I found the terminology “stitching” is not strictly defined in Section 3. The training set in line 141 is not defined. Where does it come from? The method section also does not mention how the policy uses the similarity scores. I feel the description of the method is incomplete.

The lack of clarity also makes me doubt the motivation for the use of relative representation. Actually, I do not get why changing the task representation only will enable transferring policy as it does not really affect the training of the policy. Moreover, I don’t see why relative representation of tasks will be better than PCA or domain adaptation methods like information bottleneck or MMD, let alone other representation learning methods based on generative models. The authors may need to motivate the relative representation better.

Last but not least, the zero-shot success rate is rather low. The policy seems to mostly learn to touch the object instead of solving tasks. This also makes me question the effectiveness of the approach.

**Quality Of The Limitations Section:**

Additional details required

**Questions For Rebuttal:**

I think the authors should better present and motivate their approaches. Providng clear definition and pseudocode will be of great help.

**Robotics Focus:**

Sufficient demonstration on hardware

**Summary Of Paper:**

This paper proposes a method to learn task transferable policies. The paper considers training policies for different robot arms on different tasks. Policies take task representation and robot representation as input and output actions to solve the task. A relative representation is used; after building a K-means best anchor set, it projects task information into a certain transferable representation, enabling the transferability of the robot policy. Experiments demonstrate good few-shot transfer in the simulation and also can learn transferable policy in the real world.

**Summary Of Recommendation:**

The paper studies an interesting problem. However, the writing is hard to follow. It requires a major revision to be accepted.

---

### Decision · Program_Chairs · 2023-08-30

**Decision:**

Accept (Poster)

**Comment:**

The paper proposes a novel way to learn modular policies. The idea is to put together information from different sources such as kinematics and dynamics, and task-specific things such as object states. In experiments, the approach allows transfer to different tasks and robot designs. The approach is evaluated both in simulation and on a real robot. The authors should consider expanding the experimental evaluation on a wider variety of robotic tasks.